# The Role of *TP53* in Adaptation and Evolution

**DOI:** 10.3390/cells12030512

**Published:** 2023-02-03

**Authors:** Konstantinos Voskarides, Nefeli Giannopoulou

**Affiliations:** 1Department of Basic and Clinical Sciences, University of Nicosia Medical School, 2414 Nicosia, Cyprus; 2School of Veterinary Medicine, University of Nicosia, 2414 Nicosia, Cyprus

**Keywords:** oncogene, natural selection, environment, genetics, antagonistic pleiotropy, tumor, fitness, clonal expansion, animal model, phylogenetics

## Abstract

The *TP53* gene is a major player in cancer formation, and it is considered the most important tumor suppressor gene. The p53 protein acts as a transcription factor, and it is involved in DNA repair, senescence, cell-cycle control, autophagy, and apoptosis. Beyond cancer, there is evidence that *TP53* is associated with fertility, aging, and longevity. Additionally, more evidence exists that genetic variants in *TP53* are associated with environmental adaptation. Special *TP53* amino-acid residues or pathogenic *TP53* mutations seem to be adaptive for animals living in hypoxic and cold environments or having been exposed to starvation, respectively. At the somatic level, it has recently been proven that multiple cancer genes, including *TP53*, are under positive selection in healthy human tissues. It is not clear why these driver mutations do not transform these tissues into cancerous ones. Other studies have shown that elephants have multiple *TP53* copies, probably this being the reason for the very low cancer incidence in these large animals. This may explain the famous Peto’s paradox. This review discusses in detail the multilevel role of *TP53* in adaptation, according to the published evidence. This role is complicated, and it extends from cells to individuals and to populations.

## 1. Introduction

The *TP53* gene encodes for the p53 protein, one of the main regulators of cell division and cell death. It is activated and stabilized after a variety of stresses. These stresses include radiation, toxic substances, hypoxia, high production of Reactive Oxygen Species (ROS), uncontrolled cell cycle, and others [1]. P53 can effectively trigger cell cycle arrest, DNA repair, senescence, apoptosis, and autophagy. Mutations in *TP53* are found approximately in 50% of tumors. These mutations are under positive selection since they are highly beneficial for cell survival and proliferation [2,3,4].

There is an increasing amount of knowledge related to the evolution of *TP53*. Population selective pressures on *TP53* are related to cancer, infection defense, and environmental stresses. Selective pressures at the somatic level may have a variety of effects. They can be harmful (cancer), protective, or of unknown significance. Molecular evolution research contributes to a deeper understanding of *TP53* function and generally of the wider function of tumor suppressor genes. In this review, we summarize the current knowledge about *TP53* evolution, focusing on the adaptational role of *TP53*.

## 2. Evolution of p53 Family

The gene family of *TP53* includes *TP53, TP63*, and *TP73*. The three genes share a high sequence similarity, and they act as transcription factors. Each encoded protein has three DNA binding domains [3]. The high-level conservation of p53, p63, and p73 proteins in animals reveals the significance of their function. However, plants, fungi, eubacteria, and archaea do not contain protein sequences related to the p53 family (Figure 1). Choanoflagellates have a p53 orthologue; its DNA-binding domain and C terminus are very similar to the human p53 ones [5]. The common ancestor of the choanoflagellates and humans is probably one of the first multicellular organisms, estimated to have lived 600–800 million years ago. Interestingly, animals that have evolved at later times (anemones, flies, and nematodes) all have functional proteins with high sequence similarity to the human p53 DNA-binding domains [3].

## 3. Role of *TP53* Variants in Increased Fertility and Longevity

### 3.1. Fertility and TP53

P53 family members, p53, p63, and p73, have an important role in fertility and embryonic development [6]. Here, we are going to summarize the role of p53 on fertility and reproduction, under the view of evolution and adaptation.

*TP53*-R72P is a widely studied genetic variant. It is not too clear how the 72 alleles’ frequencies are regulated by natural selection [7]. The R72 variant can induce apoptosis more effectively than the P72 one [8]. The proline allele (P) is most frequent in individuals of African descent, and the arginine allele (R) is most frequent in Caucasians. The 72 residue belongs in a proline-rich domain of p53, where a significant number of mutations has been reported [9]. A combination of alleles in this region may predispose to cancer [9].

Kang et al. (2009) [10] showed that the P72 variant is significantly enriched in an in vitro fertilization (IVF) patients’ group, especially in patients who are younger than 35 years of age. They showed that young patients, being homozygous for the P72 allele, have higher implantation and pregnancy failure rates after IVF, compared to the patients carrying at least one R72 allele. They also found evidence that the P72 effect is performed by lowering the leukemia inhibitor factor (LIF), an essential protein for embryo implantation. P53 regulates transcription of LIF through a DNA response element that is found in intron 1 of LIF gene [11]. Lledo et al. (2014) [12] found similar results. They showed that P72 is a risk factor for recurrent implantation failure and recurrent pregnancy loss. Women having the P72 allele in homozygosity have a lower rate of embryo implantation and eventually a lower pregnancy rate, in comparison with the R72 carriers [12]. Lledo et al. (2014) [12] concluded that P72 is involved in human fertility and that genotyping of the *TP53* gene could potentially determine the prognosis of IVF cycles. A meta-analysis and systematic-review study, incorporating many studies, found that recurrent pregnancy loss is associated with homozygous P72 women, most notably in Caucasians [13]. The association of the P72 allele with pregnancy loss may be population specific.

Evidence was found that the *TP53*-P72 allele may have a role after implantation, enhancing the chance for a double pregnancy. Twins’ mothers in Candido Godoi, a town in Brazil, have been found to have an increased number of P72 alleles, making the P72 a main risk factor for twinning [14]. The authors speculate that this may be related to the reduced apoptosis potential of the P72 allele. If the association of twinning with the P72 alleles is true, then the molecular mechanism needs to be deciphered.

Interestingly, there is also some evidence that relates p53 null mice with a reproduction advantage. Heterozygous p53^+/−^ male mice seem to achieve higher fertilization rates than controls. On the other hand, p53^+/−^ female mice release higher number of oocytes than controls [15]. If these observations are correct, and they are also valid in the wild animal populations, this provides a tendency for accumulation of deleterious mutations within a population, increasing carcinogenesis risk. This can be viewed under the lens of the antagonistic pleiotropy phenomenon, where mutations that are harmful at old age were selected because they are beneficial (increase the Darwinian fitness) in earlier life [16].

### 3.2. Longevity and TP53

The potential contribution of *TP53* to longevity has been studied in some animal models that will be analyzed in the next paragraphs. Evidence exists that p53 null alleles or p53 functional variants are related to longevity. The paradox is that these alleles also increase cancer risk or restrict normal p53 functions.

In *Drosophila melanogaster*, dominant negative versions of p53 in neurons extended adult life span and increased the resistance to oxidative stress in the fly [17]. Flies with a selective reduction of p53 activity in adult neurons were more resistant to paraquat poisoning than control flies. Interestingly, these flies had a normal female fecundity and a normal physical activity. These observations suggest that decreased p53 activity has positive effects on some aging phenotypes [17]. Activity regulation of Drosophila p53 seems to be like that of mammals. This regulation is performed by ubiquitination, phosphorylation, and sumoylation by multiple enzymes [18]. Another piece of evidence in *Drosophila melanogaster* is that the over-expression of p53 limits the life span in females, but favors the life span in males. On the other hand, another group showed that a p53 null mutation increased the life span of female flies, but it had less significant and more variable effects in males [19].

RNA interference or genetic knockout of the *Caenorhabditis elegans* p53 ortholog, cep-1, resulted in increased life span, depending on the daf-16 gene functionality [20]. Experiments by another group showed that a cep-1 deletion in *Caenorhabditis elegans* increased by 25% the lifespan of the worm [21]. In these experiments, the cep-1 gene was lacking the DNA-binding domain.

A study by van Heemst et al. (2005) [22] resulted in a paradox. They found that the *TP53* 72P/P genotype is associated with an increased cancer risk in humans when compared to the 72R/R genotype. However, when they analyzed 1226 people aged 85 years and over, they found that the 72P/P carriers had a 41% increased survival (*p* = 0.032), despite demonstrating a 2.54-fold (*p* = 0.007) proportional mortality risk from cancer [22]. Another study in the Danish population found an increased longevity for the 72R/P heterozygotes and 72P/P homozygous, when compared to the 72R/R homozygotes. This can be explained partly by the better prognosis after the diagnosis of cancer or other diseases. The cumulative 5-year mortality after cancer diagnosis was reduced, with an increasing number of P72 alleles. The analysis also showed that the P72 allele prolongs life on an average of 3 years [23]. Zhao et al. (2018) [24] investigated the effect of the p53 P72 allele in a mouse model. They proved that the P72 allele reduces p53 activity, increasing in this way the risk for tumor development in mice. However, mice that carry the P72 allele and escape tumor development display a longer lifespan than R72 mice and have a delayed development of aging-associated phenotypes. The authors also showed that the P72 allele influences aging and longevity through the better ability of p53 to retain and regulate the self-renewal function of stem cells, compared to the p53 R72 variant [24].

An interesting question is if Li-Fraumeni patients who carry heterozygous germ-line mutations in *TP53* and survive cancer exhibit any increased lifespan traits. No such associations have been published so far. Additionally, genetic variants of the *MDM2* gene, the main negative regulator of *TP53*, could potentially be associated with longevity. There is only one published study so far showing an association of an *MDM2* SNP with longevity, but only for female individuals [25].

## 4. *TP53* Is Crucial for Environmental Adaptation

P53 protein plays a fundamental physiological role in maintaining homeostasis by functioning as a transcription factor. P53 rapidly responds to many environmental stimuli by tetramer formation and then activation directly or indirectly a variety of genes. P53 monomers is the inactive state, and tetramers is the active state of p53 protein [26]. Its adaptive functions have an important impact on cell homeostasis, especially under changing environmental conditions caused by infections, metabolic alteration, or damage [27]. As was previously mentioned, p53 becomes activated in response to various stresses, including DNA damage, oncogene activation, oxidative stress, and hypoxia. This activation probably facilitates cellular cooperation in a coordinated manner, enabling the survival of the individual in a variety of environmental conditions. However, the validity of this assumption needs investigation.

The mole rat, *Spalax ehrenbergi*, since its origin 40 million years ago, has evolved adaptations to underground life in a highly hypoxic environment. A study revealed a special amino-acid residue in *Spalax ehrenbergi* p53 protein that is different form the conserved one in humans and mice. The amino-acid residue involves a substitution from arginine to lysine at the 174 position, in the p53 DNA-binding domain [28]. This residue is probably related to the reduced apoptosis potential of p53. This is in contrast with the fact that cancer occurrence is very rare in *Spalax* species. However, the same team showed that this Spalax p53 homologue can effectively induce autophagy, another type of cell death [29]. This altered function of p53 in Spalax is probably related to evolutionary adaptive mechanisms to hypoxia in underground life. It is very interesting that similar p53 amino-acid residues have been found in human tumors [28], this probably showing the adaptation of solid tumors in a hypoxic environment. Molecular similarities of Spalax response to hypoxia with tumor response to hypoxia are supported by published studies [30,31,32].

Zhao et al. (2013) [33] found similar results to those of Ashur-Fabian et al. (2004) [28] in some mammalian species living in a very high altitude in Tibet. Wild zokor *Myospalax baileyi* and root vole *Microtus oeconomus* have the p53 residues asparagine-104 (N104) and glutamic acid-104 (E104), respectively, differing from the serine-104 (S104) seen in other rodents, including the lowland subterranean zokor *Myospalax cansus*, and from the serine-106 (S106) in humans. More specifically, cells of *Myospalax baileyi* and *Microtus oeconomus* seem to be resistant to apoptosis under hypoxia, hypercapnia, and cold temperature [33]. These two different p53 codons are obviously an outcome of the environmental adaptation under specific ecological stresses found at the Tibet plateau. In human cancers, a germ-line mutation at this codon was reported in a patient with multiple primary cancers [34].

The subspecies *Spalax galili*, which lives in Israel, has underwent adaptive sympatric speciation due to divergent chalk and basalt ecologies of its environmental niche [35]. Higher methylation levels were noted on several sites of the p53 promoter in the population living in the chalk soil environment. In the same study, the authors showed that the diverse expression levels of p53 can affect cell-cycle arrest, but not the apoptotic mechanisms [35]. It is believed that methylation modification of p53 is an evolutionary adaptation under the environmental stresses of the abutting divergent chalk–basalt ecologies [35].

Axolotl (*Ambystoma mexicanum*, also known as Mexican salamander) is an amphibian that exhibits remarkable resistance to cancer development. Villiard et al. (2007) [36] compared axolotl p53 to the human one, and they found multiple different amino-acid residues. Interestingly, the axolotl special amino-acids have been found in human tumors [36]. Experiments showed that the activation and stability of axolotl p53 is temperature sensitive [36]. It seems that axolotl p53 variants have been selected as an adaptation to external oxygen or temperature variations.

The Andeans have adapted to the Altiplano in different ways than other natives living at high altitudes. Jacovas et al. (2018) [37] found evidence for positive selection on three genes, *CLC, SP100*, and *DUOX2*, in Native Americans living at a very high altitude in the Andean Artiplano. These genes are involved in the p53 pathway, and they are related to routes for the high-altitude hypoxia response. *CLC* encodes for a lysophospholipase that is expressed in eosinophils and basophils. *DUOX2* is involved in a p53-dependent checkpoint for cell cycle entry. *SP100* is a modulator of the p53 activity. The authors speculate that the genetic variants that are under selection on *SP100* and *DUOX2* genes contribute to a most efficient response of p53 under a hypoxic environment in high altitudes [37]. Under the same logic, the same group found that genetic variants on the *MDM2, USP7*, and *LIF* genes, encoding for proteins that are linked to the p53 transcriptional function, are probably under selection and play a significant role in the adaptation to the high altitudes of the Andean mountains [38]. Interestingly, Shi et al. (2009) [39] found that the p53 R72 allele is enriched in some East Asian human populations against the P72 allele, as an adaptation in cold winter temperatures. The authors argue that the selection of the R72 allele in cold environments is probably explained by the ability of p53 to activate mitochondrial respiration and provide more energy production during winter. Regulation of mitochondrial respiration by p53 is supported by previous studies [40]. They also found that a variant of the *MDM2* gene (mdm2 is the primary negative regulator of the p53 protein) is associated with an adaptation under high UV radiation in high altitudes [39]. We remind here that the R72 variant makes p53 more effective to induce apoptosis in comparison to the P72 variant [8].

The adaptation potential of a missense carcinogenic mutation on *TP53* gene was investigated in zebrafish, in fish larvae that were exposed under extreme starvation conditions [41]. The derived results showed that more mutated zebrafish larvae survived under extreme starvation conditions (no food at all) when compared to the wild-type ones. The experiment was performed under constant laboratory conditions, counting for larva fatalities until the 15th day post fertilization [41].

Another aspect of p53 evolution is viral infections, especially papillomavirus infections. Papillomaviruses can infect primates. Like many other viruses, papillomaviruses co-evolve with their hosts. E6 proteins of papillomaviruses stimulate polyubiquitination of p53 and subsequent proteosome-dependent degradation, thus predisposing the host for cancer [42]. Special amino-acid residues have been recognized on the host p53 and on the viral E6, being responsible for the host-pathogen-specific E6-mediated p53 degradation [43]. A study supported that non-human papillomaviruses have an adaptation toward a host carcinogenic phenotype [43]. Some macaque papillomavirus types exhibit a potential for cross-host p53 degradation, implying that complex pathogen–host interactions may exist [43].

## 5. Selection Pressures on Somatic *TP53* Mutations

### 5.1. TP53 Mutations in Normal Tissues

It is well known that cancer is formed by the expansion of specific cell populations carrying the same DNA mutations, called mutational clones [44]. Driver mutations are the ones that are primarily responsible for the transformation of normal tissues into cancerous ones. Driver mutations are under positive selection since they offer an evolutionary advantage to cells. Studies have supported the hypothesis that somatic driver mutations being under selection exists also in normal tissues, sometimes in higher frequencies than in the malignant tissues [45]. This phenomenon is not fully understood, and it shows the complexity of tumor evolution from first mutation to a benign growth and, eventually, to cancer [44]. This research was highly expedited by the recent progress of genetics technologies, especially Next Generation Sequencing. In this section, we will focus on the available evidence about *TP53* driver mutations in normal tissues.

Martincorena et al. (2015) [46] found that a surprisingly significant amount of cancer genes undergo strong positive selection in 18–32% of normal skin cells. The authors showed that aged sun-exposed skin has multiple evolving clones, and many of them carry a predisposition to cutaneous squamous cell carcinomas [46]. Interestingly, some healthy cells had 2–3 driver mutations. The same study showed that 3–5% of normal skin cells had *TP53* mutations, and strong evidence was provided for clonal expansion of p53-mutated keratinocytes [46]. 

Martincorena et al. (2018) [47] studied the normal esophageal epithelium in different age donors. It was found that a strong positive selection is acted on clones carrying mutations in genes that predispose to cancer, and these mutations accumulate with age. Surprisingly, a significantly higher amount of *NOTCH1* gene mutations was noted in normal esophageal epithelium, compared to the cancerous esophageal epithelium. *TP53* driver mutations under selection were found in 5–10% of the normal esophageal epithelium of the heathy donors [47]. Colom et al. (2020) [48] supported the above findings by finding numerous mutant clones with multiple genes under positive selection in normal esophageal epithelium, including *NOTCH1, NOTCH2*, and *TP53*. Yokohama et al. (2019) [49] also found similar results in normal esophageal epithelium, including *TP53* driver mutations under positive selection. The authors underline the fact that cancer mutations can be substantially accelerated by smoking and alcohol consumption. This excessive clonal expansion is implicated in cancer development, highlighting the importance of lifestyle factors in carcinogenesis [49].

The clonal expansions and somatic genetic changes of colorectal adenocarcinoma were also studied [50]. Some mutational processes were ubiquitous and continuous. Most mutations in colon crypts came from a single ancestral stem cell. Probable driver mutations under positive selection were present in around 1% of normal colorectal crypts. *TP53* mutations were rare among these driver mutations in normal cells, in comparison to the colorectal cancer *TP53* mutations that are found in ~56% of cases [50].

Carcinogens in cigarette smoke directly damage and mutate DNA, causing lung cancer. Yoshida et al. (2020) [51] studied the human bronchial epithelium and discovered that driver mutations increased in frequency with age, affecting 4–14% of cells in middle-aged never-smokers. In current smokers, ≥25% of cells carried driver mutations, and 0–6% cells had 2 or even 3 drivers. They used an algorithm to assess if any mutations are under positive selection in normal bronchial epithelium. Three genes were revealed through this algorithm: *NOTCH1*, *TP53*, and *ARID2*. It was clear through this study that tobacco smoking increases mutational burden and driver mutations. However, quitting smoking slows the accumulation of further damage in bronchial epithelium [51].

A study found somatic clonal expansion in morphologically normal urothelium [52]. More specifically, the chromatin remodeling genes *KMT2* and *KDM6A* were the most commonly mutated in urothelial cells. *TP53* driver mutations were found in 3.8% of the normal urothelial cells [52].

In conclusion, multiple studies show that healthy epithelial tissues accumulate cancer driver mutations with positive selection, a phenomenon that is accelerated with aging. A recent study gives some evidence why this happens. Colom et al. (2021) [53] showed that mutant clones in normal epithelium have an anti-tumorigenic role by competing and eradicating early tumors, preserving tissue integrity. Additionally, we could speculate that these mutations make tissues more resistant to harmful or challenging conditions. The continuous progress in single-cell sequencing technologies will definitely help for the better understanding of the role of somatic mutations in health, disease, aging, and evolution.

### 5.2. TP53 Mutations in Malignant Tissues

As previously mentioned, *TP53* driver mutations is a common finding in human tumors. About 50% of all tumors have mutations in the *TP53* gene [4]. However, *TP53* mutations are not always the first mutations that appear in cancer cells. It is well known for some cancer types that *TP53* mutations are selected when other cancer mutations already exist. It is not clear why this happens [3].

*TP53* mutations are commonly found together with inherited and spontaneous *BRCA1* or *BRCA2* mutations in ovarian and breast tumors [54]. Remarkably, the 70–90% of cancers with *BRCA1* mutations also have mutations in the *TP53* gene [54]. This seems to be a more general phenomenon, since *TP53* mutations frequently co-occur with mutations in DNA damage repair genes. However, the reason is still unknown.

*TP53* mutations are frequently observed in many lung, colon, and pancreatic tumors with *RAS* mutations [54,55]. Earlier studies have shown that *TP53* mutations can help *KRAS*-mutated cells to adaptively overcome replicative senescence. This may be the reason why *TP53* mutations are found together with *KRAS* mutations [56].

## 6. *TP53* Copy Numbers

Peto’s paradox is the observation that cancer incidence is not linearly associated with the number of cells of species [57,58,59]. This is a paradox since it is expected that a greater number of cells and cell divisions increases the chance of mutation accumulation and, consequently, the chance for malignancies. In 2015, Abegglen and colleagues [60] confirmed that elephants are cancer-resistant, despite their large body size and their long life span.

Abegglen et al. (2015) [60] found that elephants have at least 20 copies of *TP53*, although humans have only 1 copy. They also proved that elephants’ lymphocytes are more sensitive to apoptosis signaling in response to DNA damage from ionizing radiation when compared to the human ones. Elephants have a lower cancer rate and cancer mortality compared to humans. The multiple copies of *TP53* and the enhanced p53-mediated apoptosis could represent an evolutionary strategy for cancer suppression [60]. Consistent with this observation, transgenic mice with multiple *TP53* copies are significantly resistant to carcinogenesis [61].

Sulak et al. (2016) [62] expanded this research by including more elephant species in their study. They showed that most of the *TP53* retrogene copies of elephants are transcribed and likely translated. These *TP53* copies do not function as transcription factors, but they contribute to the enhanced sensitivity of elephant cells to DNA damage and the induction of apoptosis [62].

Padariya et al. (2022) [63] explained that elephant p53 isoforms have modified BOX-1 motifs to exhibit reduced binding capacity to mdm2 protein. P53 is regulated by the MDM2 E3 ubiquitin ligase. MDM2 is an oncoprotein and the main p53 regulator by blocking its transcriptional activity. Some species of elephants have been found to have *TP53* copies that express a variety of BOX-1 MDM2-binding motifs, enhancing the sensitivity to cellular stresses [63]. This accounts for adaptations that favor healthy aging, such as cancer defenses. Mutations altering the structures of the MDM2-p53 partners have significant functional consequences for elephants, since cancer defense can be increased when p53 isoforms escape MDM2-mediated repression [63].

Analysis of *TP53* copy numbers in multiple species can be seen in Figure 1 through a gain/loss phylogenetic tree illustration. It is obvious that most animal species have three *TP53* paralogues in their genome, most likely *TP53, TP63*, and *TP73.* A remarkable copy expansion is observed in elephants and in Ma’s night monkey.

## 7. Conclusions

There is no doubt that *TP53* promotes adaptation in many species. The adaptational role of *TP53* is summarized in Figure 2. Sound examples include the *TP53*-R72P single amino-acid substitution in humans and the multiple *TP53* copies in elephants. The obvious advantage of *TP53* selection pressures is, of course, the defense against cancer. However, evidence shows that *TP53* adaptive variants have been selected in multiple species for surviving under demanding conditions, such as cold, hypoxia, and probably viral infections, as well. Probably, there are many more adaptations to be discovered. High-throughput genetic methods will highly facilitate research of the next few years, and more intriguing aspects of the *TP53* gene will be revealed.

## Figures and Tables

**Figure 1 cells-12-00512-f001:**
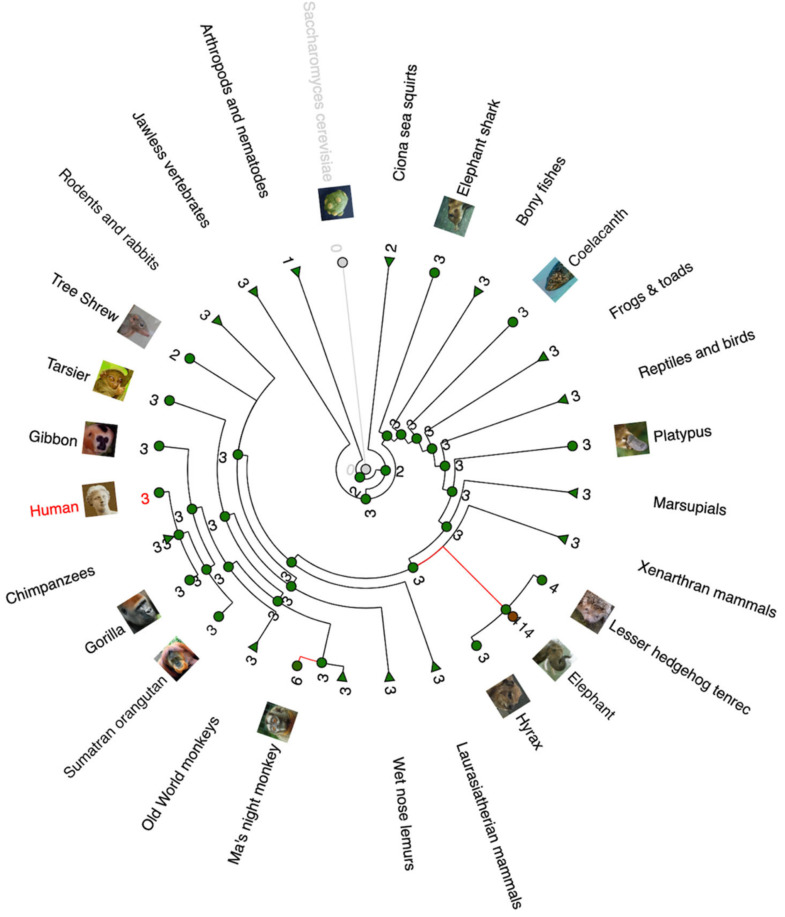
Phylogenetic gain/loss tree of *TP53* gene (created in www.ensembl.com on 17 December 2022). The tree shows the number of *TP53* paralogues/copies for each species or taxon. Red lines mark significant expansion of the *TP53* gene copies.

**Figure 2 cells-12-00512-f002:**
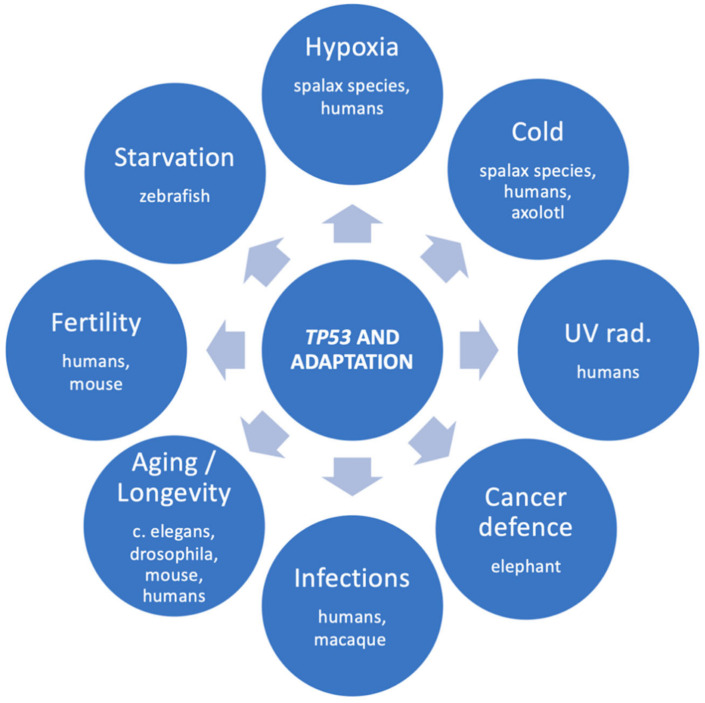
The multiple adaptational roles of *TP53* and the species that evidence was found for each role.

## Data Availability

No new data were created or analyzed in this study. Data sharing is not applicable to this article.

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
