# Peer review of "The Role of TP53 in Adaptation and Evolution"

_cells, 2023, doi:10.3390/cells12030512_

Round 1
Reviewer 1 Report
Voskarides and Giannopoulou submitted a concise review focusing on functions of the p53 transcription factor that go beyond tumor suppression. The manuscript is developed under the important lens provided by the comparison of evolutionary changes in p53 proteins and examples of apparent adaptation to particularly challenging ecological environments. One section describes recent evidence of somatic chimerism characterized by the accumulation of somatic p53 mutations in normal tissues.
The topic is interesting and worth reporting as it is less under the focus of the literature in the field. The review is generally well-written and provides, in most cases, an updated list of relevant studies. However, some sections could be significantly improved by providing more details on the reported findings, putting them into perspective evaluating if a coherent framework is emerging from the current literature on the role of p53 in organismal adaptation and which the key questions or controversies that need to be addressed next by the field. At tract, the current version of the manuscript feels just a primer or a teaser of published paper, simply pointing to them for further reading.
Below is a list of typos or expressions that need some attention and additional comments the Authors may want to consider while revising their work.
Line 10, “DNA repair”
Line 33, “Population”
Line 43, it seems that stating that the DNA binding domains of p53, p63, and p73 are almost identical may be partially misleading to readers entering the field.
Line 46, replace “that”
Line 53: “contracted”?
Line 65: the statement that proline-rich domains contain many mutation hotspots can be misleading.
Line 71: in which way does the P72 allelic variant of p53 “lower LIF levels”?
The studies cited about a distinct impact of p53 P72 vs. R72 alleles on fertility and the outcome of in vitro fertilization are a bit old. Were those studies confirmed more recently? Were those findings put in the context of the different frequencies of the two alleles in human populations?
Line 81: is there a mechanistic understanding of the possible distinct effect of P72 p53 on the frequency of twinning? It is unclear if/how the effect of p53 hemizygosity in mice on the number of activated oocytes is relevant in this context.
Section 3.2, about p53 and longevity, is interesting. Perhaps the duality between the role of p53 on longevity, cancer risk, and in particular aging-related neurodegeneration could be clarified. About the apparent paradox of P72 p53 being less tumor suppressive but associated with a longer lifespan in individuals who escape cancer formation, it could be useful to comment if this bare any relevance to the p53 heterozygous condition of Li-Fraumeni individuals or to the phenotype associated, for example, with the inheritance of a readthrough mutation in MDM2 resulting in slightly hyperactive p53.
Line 130: “responds”
Line 131: do environmental stressors lead to a more stable tetramer thermodynamic stability or a longer half-life?
Line 136: the concept that p53 activation facilitates cellular cooperation in a tissue context is interesting but requires some references to support it.
Line 145: defining autophagy as another type of programmed cell death may be misleading.
Line 147-148: it is unclear if the apparent evolutionary pressure due to the hypoxic environment in the mole rat can resemble the evolution of cancer in humans through somatic mutations.
Line 183-184: will a positive selection for variants contributing to a more efficient p53 response to hypoxia fit with the proposed model?
Line 189: which would be the mechanism that makes P72 p53 alleles more adapted to cold environment?
Section 5.1 summarizes well recent findings on the occurrence of p53 (and other cancer gene mutations) in normal tissues with aging.
Line 264: replace “along”
Line 276: replace “secondly”
Section 6 could be expanded more by considering the crosstalk between cancer risk on one side and neurodegeneration and accelerated aging on the other while referring to changes in p53 gene dosage in elephants. Perhaps it could be helpful to add to this discussion if mouse models with increased gene dosage support the view that multiple copies of p53 can represent an evolutionary strategy for effective cancer suppression.
Figure 2 is not very informative. It could be decorated by adding specific information on each highlighted connection and key references for further reading. In that case, the figure could become a quick tool to pursue a link of interest.
Reviewer 2 Report
In this review paper, the authors described the multilevel role of TP53 in adaptation. This manuscript is very interesting and very well written. On the other hand, I would like to request to add the explain at two points as below.
(1)The authors wrote “some animal models” in line 93. Please indicate the name of animals with the adequate literatures.
(2) Please indicate more detail how p53 expression is regulated in Drosophila melanogaster experiments in line 96 to line 105.
Round 2
Reviewer 1 Report
The Authors have revised the manuscript's text and figures taking into considerations the comments received. I think the revision has improved the review and the paper can be considered for publication in Cells.
Please, check Figure 2 legend.